# Rodents Prefer Going Downhill All the Way (Gravitaxis) Instead of Taking an Uphill Task

**DOI:** 10.3390/biology11071090

**Published:** 2022-07-21

**Authors:** Yehonatan Ben-Shaul, Zohar Hagbi, Alex Dorfman, Pazit Zadicario, David Eilam

**Affiliations:** School of Zoology, George S. Wise Faculty of Life-Science, Tel-Aviv University, Tel-Aviv 6997814, Israel; yonatanbs@mail.tau.ac.il (Y.B.-S.); zoharhagbi@mail.tau.ac.il (Z.H.); alex8@mail.tau.ac.il (A.D.); pazitz@tauex.tau.ac.il (P.Z.)

**Keywords:** exploration, rats, jirds, sand rats, spiny mice

## Abstract

**Simple Summary:**

In the present study, we tested whether, when given the choice to ascend or descend, rodents would favor traveling downwards or upwards on identical inclinations and sought to explain the underlying mechanism of such a preference. Our test incorporated different rodent species that dwell in different habitats and display different life and motor styles. We found that all the tested rodent species displayed a general preference to descend, with rodents from complex habitats being less affected by inclination compared with rodents from flatlands. Overall, when given the choice to ascend or descend, all the tested species displayed a preference to descend, perhaps as attraction to the ground (a behavior termed gravitaxis), where they usually have their burrows. Gravity polarizes the vertical axis and many animals can sense whether they are moving with gravity or against gravity. Gravitaxis, which is a movement in reference to a gradient, could therefore be a property involved in spatial behavior, a fundamental means of facilitating navigation, as manifested here in the rodents’ general preference to descend rather than ascend.

**Abstract:**

We directly tested whether, when given the choice to ascend or descend, rodents would favor traveling downwards or upwards. The test incorporated different rodent species that dwell in different habitats and display different life and motor styles. Testing was performed in a three-dimensional Y-maze in which the basis was horizontal and, by rotating it, one arm of the maze could be pointing upwards at a certain angle and the other arm pointed downwards at the same angle. All the tested species displayed a general preference for descent, with rodents from complex habitats being less affected by inclination compared with rodents from flatlands. Unlike laboratory rats, wild species traveled greater distances along the lower compared to the upper maze arm. All the rodents initially tended to travel the entire length of the inclined maze arms, but such complete trips decreased with the increase in inclination. When introduced into the maze from top or bottom, flatland dwellers traveled mainly in the entry arm. Overall, when given the choice to ascend or descend, all the tested species displayed a preference to descend, perhaps as attraction to the ground, where they usually have their burrows.

## 1. Introduction

Following the seminal studies of Tolman [1,2], the study of how animals, including humans, orient in space has flourished, generating substantial and extensive knowledge on spatial behavior and its controlling mechanisms. While many of these important studies have focused on traveling in horizontal two-dimensional environments, in the last two decades attention has also been devoted to understanding travel in three-dimensional environments. Some of these studies focused on flying, swimming or swinging (e.g., bats [3,4,5]; fish [6]), while others focused on surface-bounded animals that in the real world may encounter sloping terrain, bushes, rocks, trees, and other forms of vertical structures. Studying the spatial behavior of surface-bounded animals indicated that they seem to display an attraction to the lower sectors and to horizontal surfaces [7,8,9,10,11]. However, some rodents also have climbing skills that are related to their habitat requirements [12]. The burrow-dwelling Norway rat (*Rattus norvegicus*), for example, is adapted for climbing to reach food and shelters, and to avoid predators [13]; and the black rat (*Rattus rattus*) is an expert climber that nests on trees and roofs [14]. Indeed, burrowing and climbing behaviors are typical to rodents [15] and the controlling mechanism of the latter has been well-studied (e.g., [16]). The present study was aimed at direct testing of a variety of rodent species in order to determine their preference when given the choice to ascend or descend.

A common apparatus in the study of spatial behavior is the open-field—a bare horizontal platform with/without walls [17,18,19,20]. When a rodent is introduced into an open-field it soon establishes a home-base—a location in which it stays for extended periods and from which it takes excursions (round-trips) to explore the environment [21]. Home-base behavior was shown to be preserved in rats tested in large environments either alone or together with conspecifics [22], in rats that were tested in a three-dimensional lattice [7], in an open-field with a five-level pyramid in its center [8], on a cascade of bricks [10], and in an open-field that was inclined at various angles from 0° to 90° [23]. Many of the animals that were tested in these apparatuses tended to establish their home-base on the lower sectors. This preference did not depend on the entry point at which the rodents had been introduced into the apparatus [24], and even those introduced at the top soon descended to the bottom and organized their exploration from there. This preference in rats, however, differed in other rodent species. Specifically, Tristram’s jird (*Meriones tristrami*) a rodent species that resembles the Mongolian gerbil and similarly inhabits flatlands refrained from establishing its home-base at the perimeter cliffs of the apparatus and instead settled in the center [10]. When placed on a level pyramid, jirds soon descended to the floor, established there their home base, and traveled mainly on the floor or on the lower level of the pyramid [9]. In contrast, when the common spiny mice (*Acomys dimidiatus*), which dwell and forage among rocks and boulders, were placed on the level pyramid, they spent most of the time on it and seldom reached the floor [9]. This variability among different species in their response to three-dimensional environments illustrates the need for comparative studies that incorporate variation in anatomy, style of locomotion, and habitats, required in order to differentiate the general from the species-specific patterns in spatial behavior [25,26]. In the present study, we performed such a comparison, testing for a preference for either the lower or the upper sector of the apparatus in several rodent species from different types of habitats. Specifically, we compared laboratory rats (descended from the wild Norway rat, *Rattus norvegicus*) with Tristram’s jirds that inhabit flatlands, fat sand rats (*Psammomy obesus*) that forage by climbing on shrubs, and common spiny mice (*Acomys dimidiatus*) that dwell and forage among rocks and boulders. On each of these species we performed a simple test to determine whether, when given the choice, they preferred to ascend an inclined surface or descend a similarly inclined surface. We hypothesized that burrow dwellers would display a preference to descend, whereas spiny mice that are not burrow dwellers would not show such a preference.

Rodents can similarly also differ in depth and height perception and, consequently, in their behavior in three-dimensional environments. For example, common spiny mice are an exceptional genus among murid rodents (Muridae) in being precocial. They also do not dwell in burrows, but shelter in the crevices between rocks and boulders. Spiny mice also differ from rats and mice in many other aspects (see [27] for review) and they differ from flatland rodents in particular in-depth perception [28], distance perception [29], exploration [30,31,32], and excitability [33]. In accordance with such differences, it was found that environments were perceived differently even by closely related species [34,35,36,37]. In light of these considerations, we sought to determine whether laboratory rats and rodents from different habitats would choose to ascend or descend when given the choice in a vertical Y-maze.

## 2. Materials and Methods

### 2.1. Subjects

In ‘Experiment 1’, 46 Long-Evans hooded rats (weight 244 ± 46 gr), 50 fat sand rats (*Psammomy obesus*; weight 210 ± 47 gr), and 80 Tristram’s jirds (*Meriones tristrami*; weight 60 ± 15 gr) were tested, using the same number of females and males in each species. In ‘Experiment 2’, 16 female Long-Evans rats, 16 male Tristram’s jirds, and 16 male common spiny mice (*Acomys dimidiatus**;* previously considered as the subspecies *A. cahirinus dimidiatus*; weight: 41 ± 9 g; eight females and eight males) were tested. All rodents were adults (3- to 12-months-old). Rats were obtained from the animal quarters of the Faculty of Life Sciences at Tel-Aviv University and handled for 10 min daily for one week prior to testing. The other rodent species were obtained from captive colonies located at the I. Meier Segal’s Garden for Zoological Research at Tel-Aviv University. The zoo colonies are kept under natural light and temperature conditions, in cages (120 × 60 × 60 cm), each housing 5–20 females and males. The cages featured boxes, ceramic pots, and wire mesh walls on which the rodent could climb. The rationale for including these species was as follows: rats were included for comparison with other studies, since they are a common experimental animal in the study of exploration. Jirds were included since in nature they inhabit flatlands, while sand rats and spiny mice dwell in complex environments and are used to ascending and descending vertical structures. Specifically, sand rats forage by climbing on desert shrubs and feeding on their salty leaves. Spiny mice live and forage in the crevices between rocks and boulders. Accordingly, Experiment 1 constituted a comparison among rats, sand rats that are used to climbing vertical surfaces, and jirds that dwell in flatlands and are not used to climbing vertical surfaces. Due to the limited availability of sand rats, in Experiment 2 we tested rats, spiny mice that are used to climbing on vertical surfaces, and jirds that are not used to climbing vertical surfaces (details on the wild rodent species taken from [38]).

Individuals of jirds, spiny mice, and rats were kept in same-sex pairs in standard rodent plastic cages (42 × 26.5 × 18.5 cm) with a metal mesh lid and sawdust bedding. Sand rats were kept as adult female and male pairs with several juveniles in cages (27 × 32 × 47 cm) with metal mesh lid and sawdust bedding. Cages were housed for two weeks before testing in a temperature-controlled room (23 ± 1 °C) and 12/12 light/dark cycle. Fresh water and standard rodent chow were provided ad libitum, except for the sand rats, which received sugar-free rodent chow.

### 2.2. Ethical Statement

This study was carried out in strict accordance with the regulations and recommendations of the Institutional Committee for Animal Experimentation at Tel-Aviv University, which approved the protocols of the study (permits # 04-21-021 and TAU-LS-IL-2202-117-2).

### 2.3. Apparatus

The apparatus structure was adopted from Holbrook and Burt de-Perera [39]. It comprised three 65 cm long and 15 cm diameter transparent Plexiglas tubes (‘arms’) that were fused together into a Y-shape, with the arms equally spaced at 120° between them. The base of the Y-shape was placed horizontally on a 50 cm wooden base, with the base arm remaining always horizontal (‘horizontal arm’) while the two other arms could be inclined one upwards (‘upper arm’) and one downwards (‘lower arm) at the same angle (Figure 1). The far ends of the upper and lower arms were sealed with detachable plastic covers that could be removed for cleaning the arms. During testing, the horizontal arm was sealed with a transparent glass jar from which the rodent was introduced into the apparatus. In each arm of the apparatus, a rough plastic strip was attached along the lower part in order to provide the rodents with a firm grip and prevent sliding.

The apparatus was placed in a light-proofed air-conditioned room (21 ± 1 °C), illuminated by two LED lights, each equivalent to 100 w. A video camera (Ikegami B/W ICD-47 E) was attached to a rail, enabling the camera to be adjusted perpendicularly above the apparatus and capture the entire apparatus. Accordingly, the video image was always horizontal and the inclination of the arms was noted before each test.

In Experiment 2, access to the horizontal arm was blocked at the junction of the three arms of the apparatus, and the rodents could only travel along the upper and lower arms (Figure 1).

### 2.4. Procedure

In ‘Experiment 1’, the horizontal arm of the apparatus was rotated to set the other two arms at one of the following angles: 0°, 15°, 30°, 45°, or 60°. Due to the structure of the apparatus, at each of these inclinations, one arm was inclined upwards (‘upper arm’) while the other arm was inclined downwards (‘lower arm’) at the same angle (except for 0°, at which all arms were horizontal).

At each inclination in ‘Experiment 1’, groups of 10 rats and 10 sand rats were tested while jirds were tested in groups of 16 individuals/inclination. All groups included the same number of females and males. Each rodent was tested only once and only at one of the inclinations. After activating the video camera, a rodent was gently released into the far end of the horizontal arm and its behavior was recorded for 15 min. Rodents were transported to the apparatus in a transparent jar that was then gently attached to the horizontal arm, enabling them to freely explore the apparatus. At the end of the test, the rodent was released back into its cage and the apparatus was cleaned with water and detergent.

In ‘Experiment 2’, access to the horizontal arm was blocked and the two other arms were rotated to 60°. Sixteen individual rats, 16 spiny mice, and 16 jirds were tested in this inclination. Eight of each species were released into the upper arm (‘top-starters’), and eight into the lower arm (‘bottom-starters’), and each individual was tested for 10 min. All tests in both ‘Experiment 1’ and ‘Experiment 2’ were performed between 8 am and 4 pm.

#### 2.4.1. Data Acquisition and Analysis

X,Y,T coordinates were extracted from *Ethovision* at a rate of 25 frames per second for further analyses. Initial comparisons revealed no significant differences in the studied parameters between females and males in the different groups, and analyses are thus presented for both sexes together. The following parameters were processed using a script written in R software (version 4.1.0; by R-Studio team, Boston, MA, USA).

#### 2.4.2. Total Distance Traveled

The metric distances traveled by each rodent during the 15 min ‘Experiment 1’ tests or the 10 min ‘Experiment 2’ tests were recorded.

#### 2.4.3. Home-Base Location

Home-base was defined as the location in which the animal spent the longest time compared with all other location. In the present study, home-base location was referred to as the arm in which it was established. In other words, home-base location was scored as the arm in which an individual spent more time than in the other two arms.

#### 2.4.4. First Arm Entered

In ‘Experiment 1’, a rodent was introduced into the horizontal arm and had the free choice to enter the other arms. This parameter represents how many animals from each species and in each inclination entered first the upper arm and how many entered first the lower arm. An entry was considered only when all four legs were inside the arm.

#### 2.4.5. Traveling along Each Arm at Each Inclination

For each species, the total metric distance traveled along each arm was calculated at each inclination for each animal.

#### 2.4.6. Type of Trips Inside the Inclined Arms

For each entry into the upper or lower arm, the furthest point reached in that arm (= the path length in the arm) was extracted from the X,Y,T coordinates and the pool of distances underwent a Kernel Density Estimation (KDE). This is a non-parametric estimation (smoothing) of density (distribution) of path lengths. Performing KDE on each arm in each inclination and at for each species resulted in similar bimodal curves and, therefore, all data were pooled and presented as characterizing each species in all inclinations for both the upper and lower arms.

### 2.5. Statistics

Unless noted differently, one-way, two-way, or factorial analysis of variance with/without repeated measures were used to compare the different parameters, and were followed by ‘HSD for unequal N’ post-hoc test. In all tests alpha level was set to 0.05.

## 3. Results

### 3.1. Activity Decreased with the Increase in Inclination

The mean (±SEM) distance traveled in each inclination by each species and the results of a two-way ANOVA are depicted in Table 1. As shown, sand rats and jirds traveled a greater distance compared with rats. In addition, in all species, the distance traveled diminished with the increase in inclination. Altogether, the three species demonstrated a decrease in activity with the increase in inclination, regardless of their initial level of activity, which was lower in rats compared with sand rats and jirds.

### 3.2. Rodents Tended to Establish a Home-Base in the Horizontal Arm

Rodents introduced into an unfamiliar environment, like the apparatus in the present study, soon establish a ‘home-base’, which is a terminal for round-trips of exploring the environment. The surest indication of a home-base location is that the time spent there is significantly greater than in any other location. As shown in Table 2, all three species in the present experiment displayed a preference to establish the home base in the horizontal arm, where they had been introduced into the apparatus.

### 3.3. There Was an Overall Preference to Enter the Lower Arm First

After being introduced into the horizontal arm, rodents could choose to enter the upper or the lower arms. As shown in Table 3, there was an overall preference to enter the lower arm first. This preference was subtle in the shallow inclinations, but in 45° and 60° almost all rodents entered the lower arm first.

### 3.4. Traveling along the Inclined Arms Decreased with the Increase in Inclination

A comparison of the distance traveled in each of the three arms by each species for each inclination, by means of repeated-measures ANOVA, revealed a significant difference in the distance traveled in each arm (see the ANOVA results in Table 4), and a post-hoc test (HSD for unequal N) revealed that at inclinations of 30°, 45°, and 60° in all species the rodents traveled a significantly greater distance along the horizontal arm, compared with both the upper and lower arms. This difference was also echoed in the time spent in each arm by each species at each inclination (data not shown).

### 3.5. Sand Rats and Jirds but Not Rats Traveled More along the Lower Arm Than the Upper Arm

Since the significantly greater distance traveled along the horizontal arm masked possible differences between the upper and lower arms, for each species we compared only the distance traveled along the upper compared to the lower arm. In Figure 2, the distance traveled (m) by each rodent along the upper arm (ordinate) is depicted compared with the distance traveled along the lower arm (abscissa). Each data point represents the distance of one rodent at one inclination. As shown, the rats’ data points scatter close to the line of equivalence, whereas the sand rats and jirds display a notable preference to travel more along the lower compared to the upper arm. A two-way ANOVA with repeated measures revealed a significant difference between the arms in sand rats (F_1,45_ = 51.3; *p* < 0.0001) and jirds (F_1,75_ = 34.5; *p* < 0.0001) but not in rats. All three species revealed a significant interaction of inclination and upper/lower arms (rats: F_4,41_ = 17.3; *p* < 0.0001, sand rats: F_4,45_ = 3.3; *p* = 0.0218, jirds: F_4,75_ = 4.4; *p* = 0.0028). Despite the above general trends, there was also a difference among inclinations. For example, most of the rats tested at 30° (●) traveled a greater distance along the upper than the lower arm, and at 60° (○) all rats ceased to travel along both the upper and lower arms. Altogether, our findings demonstrate that whereas rats did not display an overall preference for the upper or lower arm, the sand rats and jirds demonstrated an overall preference to travel more along the lower arm.

### 3.6. Rodents Traveled Either All the Way or about Half-Way along the Inclined Arms

When traveling along the upper or lower arms, the rodents tended to turn back either at the far end of the arm or up to half-way to the end. Indeed, Kernel Density Estimation (KDE) of the maximal point reached at both the upper and lower arms revealed a bimodal distribution (two peaks). Since this characterized each arm for each inclination in all species, we first pooled the data and present them together for the definition of the two types of trips (Figure 3a). Explicit in this bimodality were two types of trips in the inclined arms: (i) **full trips**—traveling continuously to the far end of the arm; and (ii) **partial trips**—traveling about half-way or less along the inclined arm before turning back. The specific incidence of each trip type in each species and at each inclination is depicted in Figure 3b. A factorial repeated-measures ANOVA of species and inclination as the between-group factors and the incidence of full trips and short trips along the lower and upper arms as the within-group factor revealed that all factors and interactions were significant (species, F_2,161_ = 8.4, *p* = 0.0003; inclination, F_4,161_ = 18.1, *p* < 0.0001; and trip-type, F_3,438_ = 133.3, *p* < 0.0001; interactions: species x inclination, F_8,161_ = 3.4, *p* = 0.0013; species x trip-type, F_6,483_ = 21.2, *p* < 0.0001; inclination x trip-type, F_12,483_ = 30.3, *p* < 0.0001; species x inclination x trip-type, F_24,483_ = 2.1, *p* = 0.0017). As shown in Figure 3b, in all species there were initially more full trips along both the upper and lower arms, while the incidence of these full trips diminished with the increase in inclination to almost nil at the 60° inclination. Notably, in sand rats the decline in trips along the lower arm was more moderate than for the upper arm and more moderate than for both arms of the other species. In all species, there were fewer partial trips, but in rats the incidence of partial trips increased with the increase in inclination.

### 3.7. Path Visualization

The trajectories of representative individuals in each species and at each inclination are depicted in Figure 4. These trajectories illustrate the subtle effect of shallow inclinations, contrasting the decrease in travel, especially along the upper arm, at the 45° and 60° inclinations.

### 3.8. Experiment 2—The Impact of Start Points

In the above tests, rodents had the possibility to remain in the horizontal arm in which they had been introduced into the apparatus. This resulted in some the rodents refraining from traveling in the steeply inclined upper and lower arms. To prevent this possibility, we performed ‘Experiment 2’, in which access to the horizontal arm was blocked and the rodents could travel only up or down the 60° inclined arms (see Figure 1). Since we did not have sand rats that were naïve to the apparatus, we instead tested common spiny mice, which are agile rodents that dwell in a complex habitat and, like sand rats, are used to climbing up and down rocks and boulders (see Section 2.1). We therefore compared Long–Evans hooded rats both with jirds that in nature inhabit flatlands, and with spiny mice that in nature inhabit rocky habitats. Since the ‘neutral’ horizontal arm was blocked and not available as a start point, two groups were tested in each of the three species. In one group, each rodent was introduced into the apparatus from the upper arm (‘top-starters’), while in the other group, each rodent was introduced from the lower arm (‘bottom-starters’). Both arms were inclined at 60°. Figure 5a depicts the distance traveled by each individual rodent in the upper compared to the lower arm. As shown, rats and spiny mice scatter near the line of equivalence with some preference of bottom-starters to travel a greater distance along the lower arm. Jirds, however, displayed a strong preference for the arm in which they had been introduced into the apparatus: all bottom-starter jirds except for one traveled a greater distance along the lower arm, whereas all top-starters jirds but one did not reach the lower arm and restricted their travel to the upper arm. Indeed, a factorial ANOVA of the distances traveled by top-starters and bottom-starters in each species in the upper compared to the lower arm revealed an effect of species (F_2,42_ = 22.3; *p* < 0.0001), no significant effect of the start point (top or bottom; F_1,42_ = 0.04; *p* = 0.8469), and a significant interaction of species x start point (F_2,42_ = 3.4; *p* = 0.0417). There was no significant effect of the arm (upper or lower) and no interaction of arm × species (F_1,42_ = 0.6, *p* = 0.8141 and F_2,42_ = 0.7, *p* = 0.5217, respectively), but the interaction of arm (upper or lower) × start point and the interaction of arm × species × start point were significant (F_1,42_ = 17.7, *p* = 0.0001 and F_2,42_ = 4.1, *p* = 0.0234, respectively). A post-hoc Tukey HSD test revealed a significant difference between the distance traveled by jirds along the upper compared with the lower arm, but no such significant difference in the two other species.

The preference of top-starters for the upper arm and the preference of bottom-starters for the lower arm were also apparent in the time they spent in each arm. As shown in Figure 5b, the rodents divided the 10 min of the test between the two arms but with a general preference for the starting arm. This division was most drastic in jirds, in which all but one in each group remained in the starting arm throughout almost the entire 10 min test. The videoclips revealed that top-starting jirds remained ‘stuck’ for relatively long periods at the top of the upper arm like a cat up a tree. Altogether, these results indicated that while there was a preference for traveling along the lower arm, there was also an impact of the particular starting point. In jirds, the impact of starting from the top was conclusive, and they remained mostly at the point of entry, with brief journeys along the upper arm.

A.In rats, the distance traveled by top-starters and bottom-starters scattered near the line of equivalence, indicating little difference between top- and bottom-starters. This was also the case with spiny mice which are used to ascending and descending in their natural habitat. In contrast, in jirds, which inhabit flatlands, all but one bottom-starter traveled a greater distance along the lower arm while all but one top-starter traveled solely along the upper arm and did not enter the lower arm.B.In rats and spiny mice, there was a slight trend of top-starters to spend more time in the upper arm, and of bottom starters to spend more time in the lower arm. This became very distinctive in jirds, with all but one bottom-starter spending more time in the lower arm while all but one of the 10 top-starters spent all the time in the upper arm and never entered the lower arm. Moreover, these latter jirds remained at the far top end of the upper arm, where they had been introduced into the apparatus, and remained there for the entire test duration.

The above-noted results of top-starters and bottom-starters are illustrated in Figure 6. The trajectories of two exemplary rodents from each group are depicted, illustrating that while both top- and bottom-starter rats and spiny mice traveled along both arms, top-starting jirds traveled only in the upper arm.

## 4. Discussion

We compared the behavior of different rodent species from different habitats and displaying different motor styles: laboratory rats, which are a common experimental model, kept in a dull environment that does not conduce to various activities; jirds that naturally inhabit flatlands over which they gallop; and sand rats or spiny mice, which are used to ascending and descending while traveling and foraging in their complex habitats. When individuals of these species were given the choice of going up or down along an inclined arm of a y-maze, they preferred to first enter the lower arm rather than the upper arm, as hypothesized. Most of their traveled distance took place along the horizontal arm, where they typically established their home-base. However, a comparison of only the distance traveled along the upper arm with that traveled along the lower arm revealed that both the sand rats and jirds tended to travel more along the latter than along the former, whereas the rats traveled about the same distances along both upper and lower arms. The preference of rats and spiny mice for the lower arm was also evident when access to the horizontal arm was blocked. However, unlike the rats and spiny mice, the flatland-dweller jirds that were tested with the blocked horizontal arm restricted their travel to the arm through which they had been introduced into the apparatus. This was especially notable in jirds that had been introduced into the upper arm, where they remained for long periods at the top like a cat trapped up a tree. Therefore, while we hypothesized that burrow dwellers would favor descending, the top-starter jirds in Experiment 2 descended only half the way, perhaps due to the steep inclination of 60°.

Animals differ in their motor abilities and in possessing sensory organs with different sensitivities, and such differences can directly influence their three-dimensional behavior [26]. This has long been noted in the study of depth perception, which constitutes a particular perspective of the vertical dimension and seems to be linked to the structure of an animal’s habitat and life style. Mongolian gerbils resemble Tristram’s jirds: both belong to the genus *Meriones* and both inhabit grasslands and prairies. Depth perception in Mongolian gerbils was found to differ from that in spiny mice that live and forage in crevices among rocks and boulders [28,29]. This difference in depth perception between flatland dwellers (*Meriones* spp.) and those dwelling in complex habitats may account for the difference we found between the jirds, which represent flatland dwellers, and the spiny mice and sand rats which dwell in complex habitats and are accustomed to vertical ascent and descent. Interestingly, in contrast to the difference in their behavior in the vertical domain, past studies have revealed no difference between flatland and complex-environment dwellers in their distance perception, and they all performed equally when jumping over gaps (for gerbils [40,41,42]; for spiny mice [29]). The lack of difference in distance perception could also account for the present findings, which revealed only minor differences among species in terms of the traveled distance. The difference in perceiving the vertical domain, together with the lack of such a difference in distance perception, further supports the notion regarding a differential perception of the vertical and horizontal dimensions (the ‘bi-coding model’, [25]).

The present findings unequivocally demonstrate that when given the choice, rodents prefer to descend rather than ascend. In a previous study, rats that were offered a choice between ascending and descending, for both of which they obtained the same reward, preferred to descend [43]. In both the two latter studies, the bias towards lower sectors was explained in terms of energy conservation. Similarly, we found that sand rats and rats that dwell and forage in complex habitats were also biased towards the lower sectors [8,10]. This differed, however, in rocky habitat dwellers, the spiny mice, which preferred the upper sectors [9], whereas the flatland-dweller jirds preferred lower sectors and moved away from cliffs and upper sectors [9,10]. The bias towards descent rather than ascent has usually been interpreted in terms of energy cost. Nevertheless, it has also been suggested that ascending/descending involves psychological components too, such as fear of falling or seeking safety (see [10,25] for review). These factors could account together for the present finding of a preference to descend rather than ascend. Altogether, while the bias for lower sectors in the above-noted studies was a noted as a part of general spatial behavior, in the present study it was directly demonstrated as the preferred free choice.

In a seminal review on three-dimensional spatial behavior [25], it was suggested that behavior in the vertical domain should be studied from four different perspectives: (i) processing of verticality cues; (ii) navigation on a sloping surface; (iii) navigation in multilayer environments; and (iv) navigation in a volumetric space. In this context, the present study reflects the first perspective—of processing verticality cues; namely, gravity. Indeed, gravitaxis, like other forms of taxis, is a fundamental means of facilitating navigation. Gravity polarizes the vertical axis and many animals can sense whether they are moving with gravity (geotaxis) or against gravity (negative geotaxis). Notably, it was suggested that the phylogeny of the ability to form a mental representation of the environment (cognitive mapping) comprises an integration of two mapping systems. The more ancestral system originated in marine animals (where life began) in the form of a bearing (directional) mapping based on gradients. This preceded the second system—that of a sketch (positional) mapping of landmarks, which developed with the move to terrestrial life [44]. Gravitaxis, which is a movement in reference to a gradient, could therefore be an ancestral property involved in spatial behavior, as manifested here in the rodents’ general preference to descend rather than ascend.

Exploration can also be dominated by an urge to optimize security. Specifically, it was suggested that when animals are unmotivated to forage and have nothing to gain in terms of food, water, etc., the default is to seek a safe place to stay [45]. For rodents that are burrow dwellers, like rats, sand rats, and jirds, attraction to the bottom could derived from the urge to reach a safe haven, i.e., their burrows in the ground [7,8,10]. All rodents in the present study were unfamiliar with the environment (apparatus) into which they were introduced, and their natural response in such events is exploration. Several theories (e.g., [46,47,48]) have suggested that the novelty of an unfamiliar environment induces anxiety or fear, and the consequent exploration, which results in familiarity with the environment, reducing uncertainty and anxiety. Gravitaxis in an unfamiliar environment could represent an attraction to where a shelter (burrow) would usually be located, and therefore fits well into exploration as a behavior aimed at reducing anxiety.

## 5. Conclusions

The need for comparative studies of spatial behavior in three dimensional environments has been noted in key reviews of this field of research (e.g., [25,26]). The present study, which compared several species of wild rodents and laboratory rats, was aimed at a specific question: would rodents from different habitats, with different life styles and movement styles, prefer to ascend or descend when given the choice. All the tested species displayed a general preference for descent, with the rodents from complex habitats being less affected by the degree of inclination compared with the rodents from flatlands. In all the species, activity diminished with the increase in inclination, perhaps due to both the need for greater physical effort and to psychological factors such as fear of falling and seeking safety. Compared with the behavior of the wild species, the impact of inclination on the laboratory rats was more subtle, perhaps since for many generations these are reared in a dull environment that does not stimulate rich movement styles. Rats are also larger in terms of body mass, and this could also have affected their behavior in the present study compared with the smaller and more agile wild species. Despite the study’s findings demonstrating diversity in the behavior of the different species in the vertical dimension, when given the choice to ascend or descend, all the species displayed a general bias towards descent.

## Figures and Tables

**Figure 1 biology-11-01090-f001:**
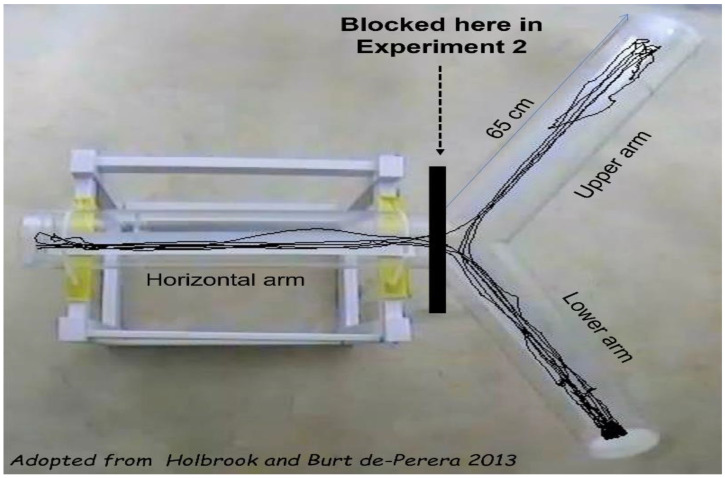
The apparatus. Three 65 cm transparent Plexiglas tubes (diameter: 15 cm) were connected, spaced planarly at 120°. One arm was attached horizontally to a wooden construction and could be rotated (‘horizontal arm’). In ‘Experiment 1’, rotation angles were: 0°, 15°,30°, 45°, and 60°. In each such rotation angle, one arm was inclined upward (‘upper arm’) and the other was inclined at the same angle downward (‘lower arm’). The upper and lower arms were sealed at the far end with a detachable plastic cover. The far end of the horizontal arm was the inlet at which the rodents were introduced to the apparatus by means of a transparent jar that remained there throughout the test. In ‘Experiment 2’, the arms were rotated to 60° and access to the horizontal arm was blocked at the arms’ junction by means of plastic cover. In this experiment rodents were introduced into the apparatus from the far opening of the upper arm (‘top-starters’) or the far end of the lower arm (‘bottom-starters’).

**Figure 2 biology-11-01090-f002:**
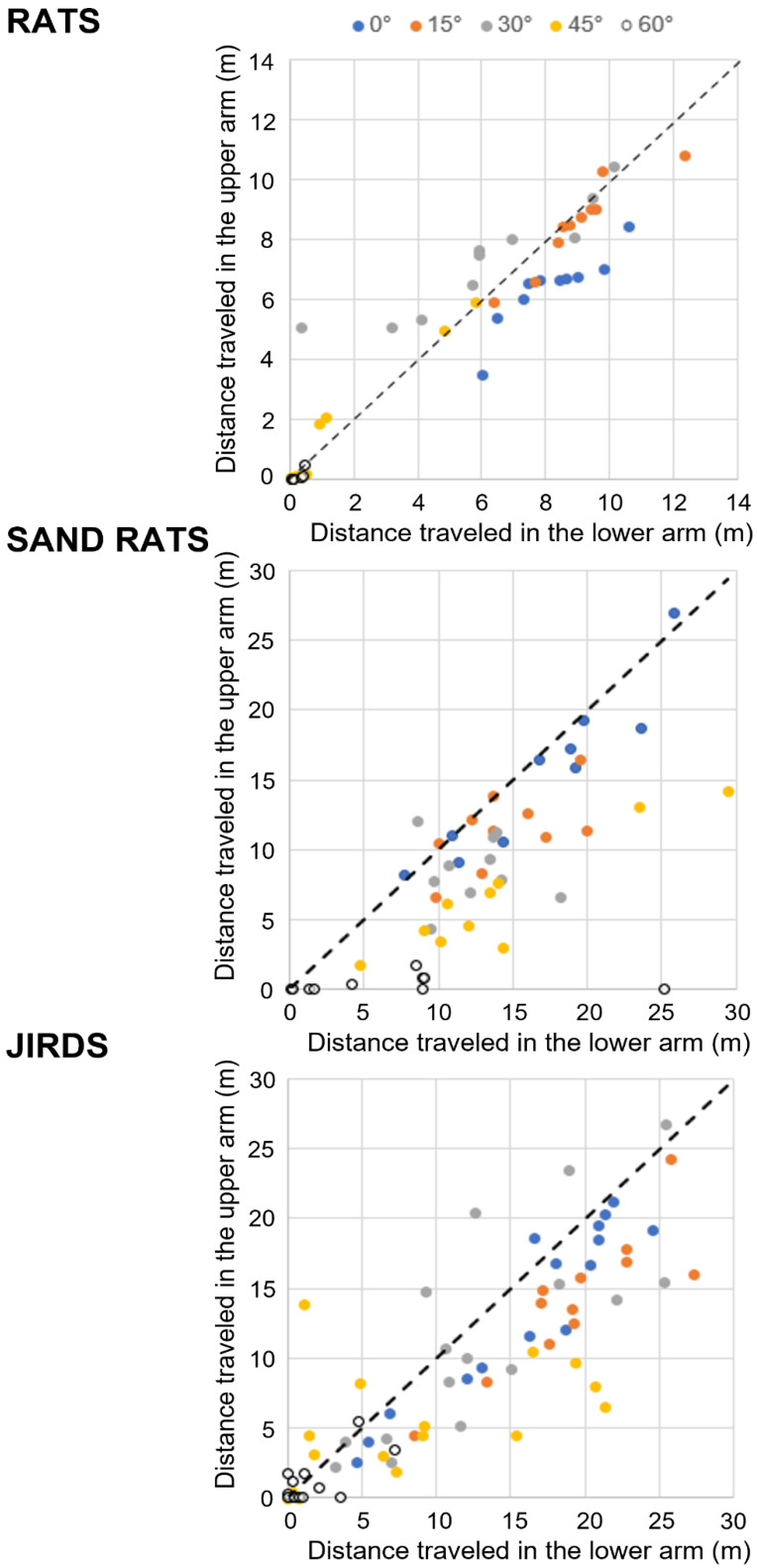
Distance traveled in the upper vs. the lower arm. In each inclination in each of the three species, the distance travelled by each rodent in the upper arm (ordinate) is plotted against the distance traveled by the same rodent in the lower arm (abscissa). Accordingly, for each species, each dot represents one rodent in one inclination. The various inclinations of the upper and lower arms are depicted at the top of the figure. The diagonal dashed line is the equivalence line along which the distance traveled in both arms was identical. As shown, rats aggregated near the line of equivalence, although those tested in 0° were blow the line (traveled a little more in the lower arm), those tested in 15° were on the line of equivalence, those tested in 30° were above the line (traveled a little more in the upper arm), while those tested in 45° and 60° ceased to travel in both arms. While the diminished travel in the higher latter inclinations is also apparent in jirds and sand rats, in the shallower inclination almost all sand rats and jirds aggregated below the line of equivalence (traveled greater distances in the lower arm).

**Figure 3 biology-11-01090-f003:**
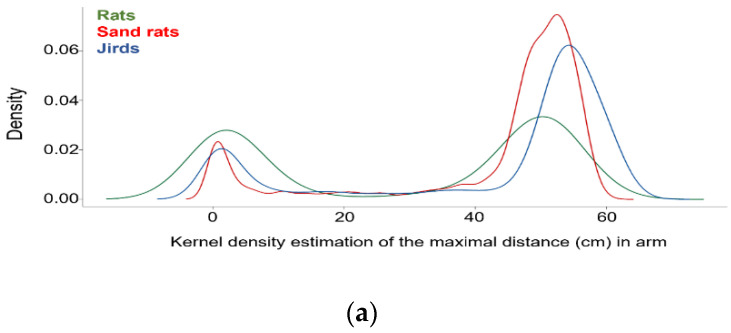
(**a**) Kernel Density Estimation of the maximal distances traveled in each entry to the upper and the lower arms, by each of the three species. The abscissa is a smoothing of the distances traveled in each entry to either the upper or lower arms, representing roughly the length of the arms (60 cm). Note that the X,Y,T coordinates that were extracted from Ethovision refer to the center of the body image of the rodents and therefore 0 on the abscissa represents about half the body inside one of the arms, and 50 represents the arrival to the far end of the arm. The ordinated represents the smoothen distribution of the incidence of the length of entering the arms. As shown, distribution in all three species was bimodal, with either very short entry to the arm (partial trips), or entering the arms all the way to their end (full trips). (**b**) Mean (± SEM) of full trips and partial trips for each species and each inclination of the upper and lower arms. * indicates a significant difference in Dunnet’s test, compared with the 0° inclination. As shown, in all species and in both the upper and lower arms there were more full than partial trips, but the incidence of the former decreases with the increase in inclination. The sand rats were exceptional, showing a significant decrease only at the 60° inclination. In all species, partial trips increased with the increase in inclination, especially apparent along the lower arm and in rats.

**Figure 4 biology-11-01090-f004:**
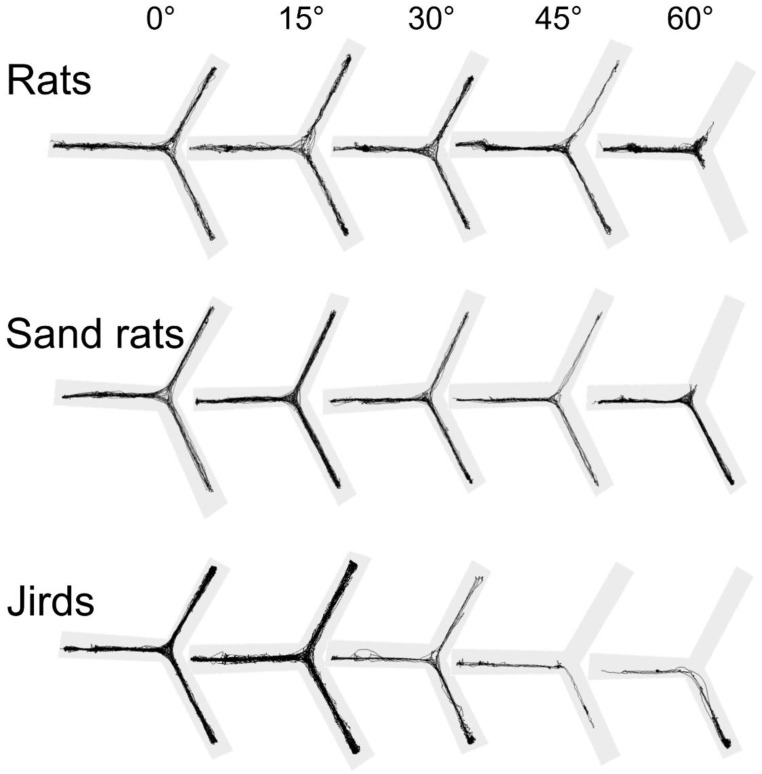
Exemplary trajectories of individuals of each species at each inclination. As shown, changes were subtle at the shallow inclinations of 15° and 30° compared with 0°. At the steep inclinations of 45° and 60°, the rats ceased to travel along the inclined arm, while the sand rats and jirds also did not enter the upper arm, but also spent less time in the lower arm.

**Figure 5 biology-11-01090-f005:**
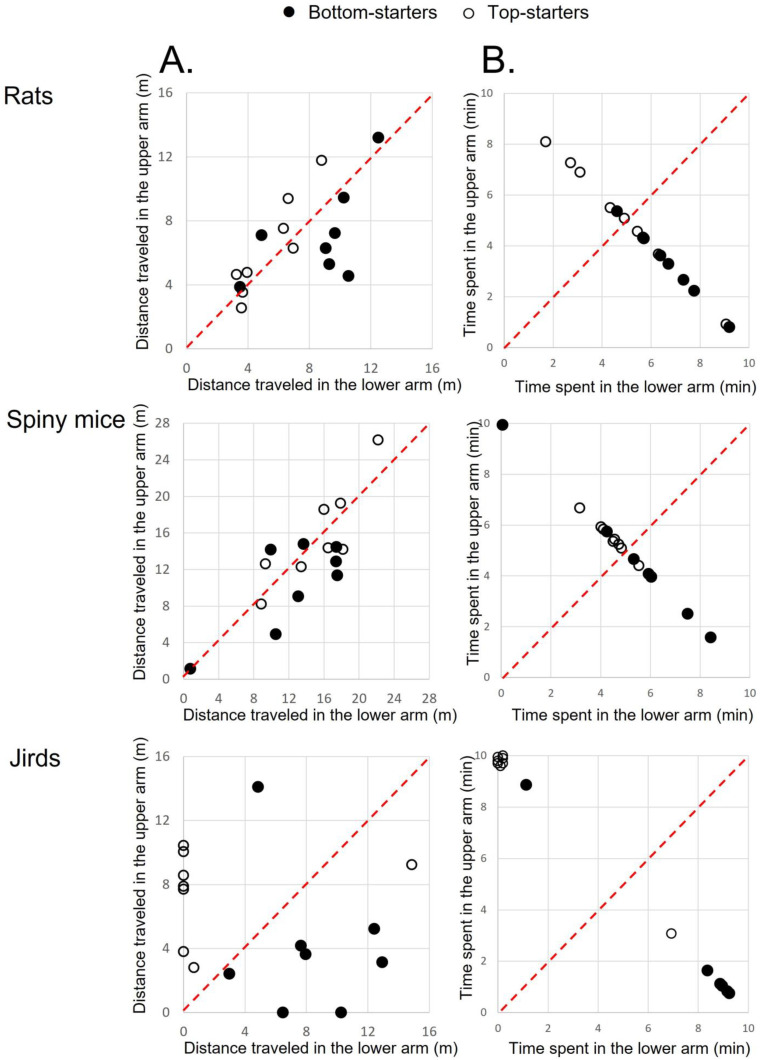
The distance (m) traveled (**A**) and the time (min) spent (**B**) along the upper arm (ordinate) vs. the lower arm (abscissa) in the three species is depicted for top-starters and bottom starters. The dashed line represents a similar distance or time spent on both arms.

**Figure 6 biology-11-01090-f006:**
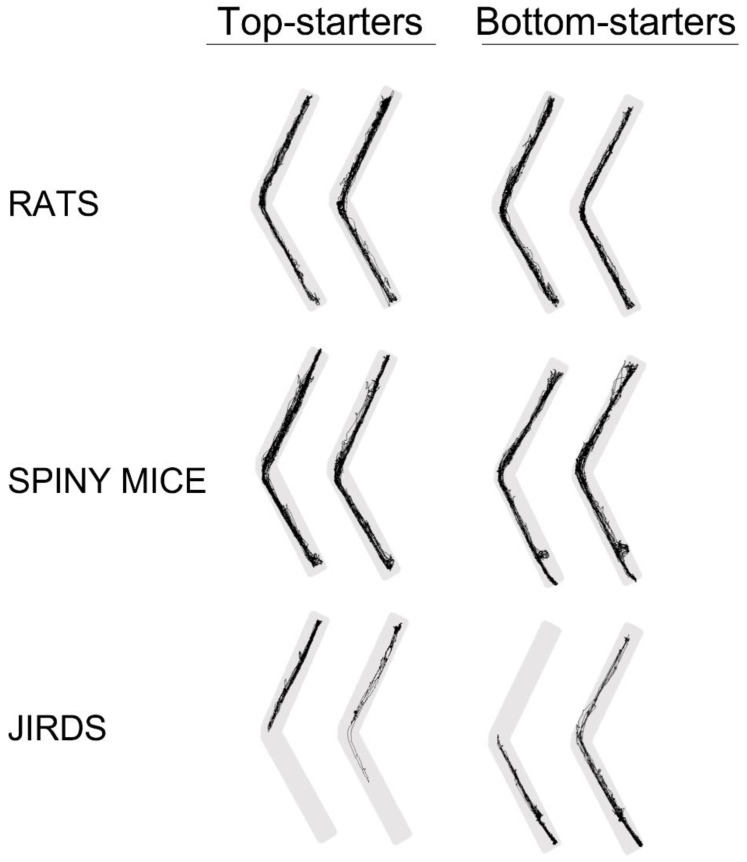
Trajectories of two individual ‘top-starters’ and two ‘bottom-starters’ of each species. As shown, both rats and sand rats traveled along the upper arm regardless of their start point, whereas ‘top-starter’ jirds traveled only along the upper arm and ‘bottom-starter’ jirds traveled mainly along the lower arm.

**Table 1 biology-11-01090-t001:** Mean (±SEM) of the total distance (m) traveled by each species in each inclination. Results of a two-way ANOVA are depicted at the bottom of the table. In each species, a significant difference in each inclination, as revealed in a post-hoc comparison to 0° inclination by Dunnet test is depicted by * and bold font. Note: in the rats tested at 60°, four rats did not move at all after being introduced into the apparatus, and were therefore removed from further analyses.

	Distance Traveled (m)		
Inclination	Rats	Sand Rats	Jirds
0°	30.1 ± 1.9	59.9 ± 6.5	58.1 ±5.6
15°	31.0 ± 1.7	47.3 ± 3.4	69.9 ± 5.4
30°	26.4 ± 1.9	**40.4 ± 2.1 ***	47.5 ± 5.8
45°	**19.3 ± 1.9 ***	44.8 ± 6.6	**33.7 ± 5.5 ***
60°	**15.0 ± 1.4 ***	**33.2 ± 3.0 ***	**26.9 ± 4.1 ***
Effect of Species		F_2,161_ = 31.1	*p* < 0.0001
Effect of Inclination		F_4,161_ = 13.6	*p* < 0.0001
Interaction of Inclination × Species		F_8,161_ = 2.7	*p* = 0.0084

**Table 2 biology-11-01090-t002:** The percentage of rodents that established their home-base in each of the arms. The number of rodents in each group is depicted beside the name of each group. For each arm at each inclination, the proportion of animals that established the home-base in that arm is depicted. As shown, there was a strong preference to establish the home-base in the horizontal arm. While at shallow inclinations there were a few rodents that established the home-base at the far end of the inclined arm, in the steep inclination of 45° and 60°, the home-base was almost exclusively established in the horizontal arm.

	Inclination	Horizontal Arm (%)	Lower Arm (%)	Upper Arm (%)
Rats (*n* = 10, except for 60° where *n* = 6)	0°	90	10	0
	15°	60	10	30
	30°	70	10	20
	45°	100	0	0
	60°	100	0	0
Sand rats (*n* = 10)	0°	70	10	20
	15°	50	30	20
	30°	70	30	0
	45°	90	10	0
	60°	90	10	0
Jirds (*n* = 16)	0°	44	25	31
	15°	25	56	19
	30°	56	31	13
	45°	75	19	6
	60°	88	12	0

**Table 3 biology-11-01090-t003:** The percentage of entries into the lower and upper arm first. Group size is depicted beside each species. The percentage of individuals that progressed to enter the lower or the upper arm first is depicted for each inclination in each species. Note that the group of jirds was larger than those of rats and sand rats.

Species	Inclination	Entries to the Lower Arm (%)	Entries to the Upper Arm (%)
Rats (*n* = 10; except for 60° where *n* = 6)	0°	70	30
	15°	60	40
	30°	80	20
	45°	10	90
	60°	100	0
Sand rats (*n* = 10)	0°	60	40
	15°	50	50
	30°	50	50
	45°	100	0
	60°	100	0
Jirds (*n* = 16)	0°	56	44
	15°	69	31
	30°	75	25
	45°	75	25
	60°	100	0

**Table 4 biology-11-01090-t004:** The mean (±SEM) of the distance traveled along each arm by each species at each inclination. Results of the ANOVA are depicted below the means, demonstrating a significant effect of arm, inclination, and species, as well as significance in all interactions among these factors. * indicates that the distance traveled along the horizontal arm at this level is significantly greater than the distance traveled along both the upper and lower arms (HSD test for unequal N).

		Distance Traveled in Each Arm (m)		
	Inclination	Horizontal	Lower	Upper
Rats	0°	15.5 ± 1.2	8.1 ± 0.5	6.4 ± 0.4
	15°	13.4 ± 0.9	9.0 ± 0.5	8.5 ± 0.5
	30°	13.1 ± 0.7 *	6.0 ± 1.0	7.3 ± 0.6
	45°	16.4 ± 1.2 *	1.4 ± 0.7	1.5 ± 0.7
	60°	14.6 ± 1.3 *	0.2 ± 0.1	0.1 ± 0.1
Sand rats	0°	27.8 ± 2.9	16.8 ± 1.9	15.3 ± 1.9
	15°	21.4 ± 1.6	14.5 ± 1.2	11.4 ± 0.9
	30°	19.4 ± 1.1 *	12.4 ± 1.0	8.6 ± 0.8
	45°	24.2 ± 3.1 *	14.1 ± 2.4	6.5 ± 1.4
	60°	26.0 ± 1.2 *	6.8 ± 2.5	0.4 ± 0.2
Jirds	0°	26.6 ± 2.3	17.0 ± 1.9	14.6 ± 1.9
	15°	31.1 ± 2.0	22.6 ± 1.9	16.1 ± 1.4
	30°	22.5 ± 2.6 *	13.3 ± 1.9	11.7 ± 2.1
	45°	20.0 ± 3.3 *	8.5 ± 2.1	5.3 ± 1.0
	60°	24.5 ± 3.9 *	1.5 ± 0.6	0.9 ± 0.4
Effect of Species			F_1,161_ = 31.1	*p* < 0.0001
Effect of Inclination			F_1,161_ = 13.6	*p* < 0.0001
Interaction of Inclination × Species			F_1,161_ = 2.7	*p* = 0.0084
Effect of Arms			F_2,322_ = 513.6	*p* < 0.0001
Interaction of Arm × Species			F_4,322_ = 8.3	*p* < 0.0001
Interaction of Arm × Inclination			F_8,322_ = 15.4	*p* < 0.0001
Interaction of Arm × Species × Inclination			F_16,322_ = 2.0	*p* = 0.0142

## Data Availability

Data available upon request.

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
