# Peer review of "Rodents Prefer Going Downhill All the Way (Gravitaxis) Instead of Taking an Uphill Task"

_biology, 2022, doi:10.3390/biology11071090_

Round 1
Reviewer 1 Report
This paper has an appropiate research design, methods are adequately described and the results clearly presented. For these reasons I recommend the paper to be published by Biology in its present form
Author Response
Comment: This paper has an appropriate research design, methods are adequately described and the results clearly presented. For these reasons I recommend the paper to be published by Biology in its present form
Reply: Thanks for your kind words
Reviewer 2 Report
The paper tests whether, when given the choice to ascend or descend, rodents would favor traveling downwards or upwards.
The introduction is clear and complete; the section on materials and methods reports in detail the various phases of the experiment.
The results are also very interesting and can be useful for protecting the welfare of laboratory animals and other rodents kept in unnatural environments. I only have a few requests to make the figures easier to read.
Table 3 shows the percentage of individuals who progressed to enter the lower or the upper arm first and not the number of entries into the lower and upper arm first. I think it is better to correct the text.
In table 4 and other parts of the text, the abbreviation for meters in my opinion should be (m) and not (m.).
Figure 2 reports the distance traveled in the upper vs. the lower arm: it would be useful to insert the legend on the axes: for example distance traveled (m).
In the figure 3a I would add the unit of distance measurement (m) on the x-axis.
In figure 5 I would modify the writing on the x and y axes by adding that it is the distance (m).
I have no other suggestions to make and I believe the paper can be accepted for publication with these small changes.
Reviewer 3 Report
The literature well supports the introduction, and the work is interesting.
I would suggest including a hypothesis.
Line 28 avoids the space in the parenthesis,
Line 82, substitute "animals" for rodents. It is more accurate
Line 108, 124, please put the measures of the cages in parenthesis. To make a homogenous text.
Line 120, avoid putting a period before the parentheses
Line 135, it isn't very clear if the apparatus arms measure 65 cm or 60 cm (as you show in figure 1)
Line 154, Figure 1. Please consider that it is better for the reader if you describe in this part only the apparatus because it is repetitive reading, almost the same as the procedure
Line 191, Please consider removing this paragraph and explain the time that you consider for each experiment in procedures
Line 196, you mention that "the apparatus was divided into sectors," but this is unclear if you refer only to the "arms" or if each "arm" has its sector
Line 200, Please consider removing this paragraph and explain the time that you consider for each experiment in procedures at experiment 1
Line 205, Please consider mentioning the units
Line 230, is the total distance measured in meters? Could you please clarify?
Line 246, Table 2, please specify if this table refers to the "Percentage home base in each of the arms." Please remove each percentage (%) from each result.
Line 284, Table 3, the result at 45º in the upper arm should be 90%. You can add at the top of "Lower arm" and "Upper arm" the symbol of % (avoiding adding it to each result)
Line 295, Table 4, avoid the line (4._). Please check the manuscript the values of SEM at the table are not on the same line.
Line 447 and 448, please fill the parenthesis with the symbol
In the discussion, line 518, you mention that rats obtain the "same reward" did you give food or another rewarding stimulus? or refer to the choice of ascending or descending?
Have you considered that using a transparent apparatus with light in the environment could influence the preference for being at the bottom? As you mention in line 557, this might cause more anxiety?
How do you think could influence the schedule (8 am to 4 pm) that you used for testing the individuals in the task? I mean, in terms of motivation (p.ej. hunger)
Thank you for your effort and time.
Round 2
Reviewer 3 Report
I consider that the text is now clear and understandable. You did nice work improving the document and doing the clarifications.